# Tool Wear State Recognition Based on One-Dimensional Convolutional Channel Attention

**DOI:** 10.3390/mi14111983

**Published:** 2023-10-26

**Authors:** Zhongling Xue, Liang Li, Ni Chen, Wentao Wu, Yuhang Zou, Nan Yu

**Affiliations:** 1Key Laboratory of Advanced Manufacturing and Intelligent Technology, Ministry of Education, Harbin University of Science and Technology, Harbin 150080, China; 2College of Mechanical and Electrical Engineering, Nanjing University of Aeronautics & Astronautics, Nanjing 210016, China; 3Institute of Materials and Processes, School of Engineering, University of Edinburgh, Edinburgh EH8 9BT, UK

**Keywords:** tool condition monitoring, tool wear state recognition, 1D Convolution, channel attention

## Abstract

Tool wear state recognition is an important part of tool condition monitoring (TCM). Online tool wear monitoring can avoid wasteful early tool changes and degraded workpiece quality due to later tool changes. This study incorporated an attention mechanism implemented by one-dimensional convolution in a convolutional neural network for improving the performance of the tool wear recognition model (1DCCA-CNN). The raw multichannel cutting signals were first preprocessed and three time-domain features were extracted to form a new time-domain sequence. CNN was used for deep feature extraction of temporal sequences. A novel 1DCNN-based channel attention mechanism was proposed to weigh the channel dimensions of deep features to enhance important feature channels and capture key features. Compared with the traditional squeeze excitation attention mechanism, 1DCNN can enhance the information interaction between channels. The performance of the model was validated on the PHM2010 public cutting dataset. The excellent performance of the proposed 1DCCA-CNN was verified by the improvement of 4% and 5% compared to the highest level of existing research results on T1 and T3 datasets, respectively.

## 1. Introduction

The high-speed contact between the cutting edge and the workpiece surface during the cutting process leads to increasing tool wear, which can affect the machining accuracy [1]. In particular, excessive tool wear can lead to waste due to unmet manufacturing requirements for the workpiece [2]. Tool wear state recognition allows real-time monitoring of tool wear for tool replacement and maintenance to avoid production interruptions and cost losses due to severe tool wear [3]. Therefore, it is important to realize the tool wear state recognition.

Tool condition monitoring (TCM) tasks are mainly categorized into direct and indirect methods. The direct method directly observes the wear area of the tool and measures the tool wear through the microscope, which provides more intuitive and accurate results. With the advancement of intelligent algorithms, many researchers are currently realizing the segmentation of the tool wear area by means of machine vision to achieve the measurement of tool wear [4,5,6]. However, the direct method requires stoppage for tool observation, which reduces the efficiency of machining. Moreover, chips and cutting fluids affect the observation of tool wear areas. Therefore, the direct method is not suitable for actual machining. The indirect method is realized by monitoring cutting signals such as force signals [7], vibration signals [8], and acoustic emission signals [9] to monitor the tool condition. These signals are collected in real time by sensors mounted on the workpiece or spindle, etc., which have less influence on the machining process and are used in more application scenarios. Therefore, it is important to study the mapping algorithm of cutting signal and tool state for TCM system.

With the development of artificial intelligence technology, more and more intelligent algorithms are applied to tool wear monitoring tasks [10]. Qin et al. [11] used stacked sparse self-coding networks for tool wear monitoring. Duan et al. [12] converted cutting signals into time-frequency maps by short-time Fourier transform, then feature extraction by PCANet and GA-SPP, and finally tool wear prediction by SVM. Wei et al. [13] screened the sensitive low-dimensional features of force signals by whitening variational mode decomposition (WVMD) and Joint information entropy (JIE), and optimal-path forest (OPF) was used as a classifier to realize the tool wear state recognition. Chan et al. [14] achieved tool wear state monitoring by extracting overall and local features of the signal through LSTM. Zhou et al. [15] utilized graph neural networks to achieve tool wear state monitoring with small samples. Hou et al. [16] used lightweight networks to achieve tool wear state monitoring after augmenting and balancing unbalanced data via WGAN-GP. These intelligent algorithms achieved excellent results in the task of tool wear monitoring.

Attentional mechanisms can enhance the perceptual ability, adaptability and interpretability of neural networks [17]. With the use of a large number of sensors, the channel dimensions of cutting signals are increasing, and the attention mechanism can effectively combine data from different channels to improve the performance and interpretability of the model. Therefore, more and more researchers are applying attention mechanisms to tool wear monitoring tasks. Li et al. [18] transformed the force signal into a time-frequency map by continuous wavelet transform and established a channel space attention mechanism to realize tool wear state monitoring. Zeng et al. [19] fused and converted the multi-sensing data into images, and selected the information in the channel and spatial domains through the attention mechanism to realize the deep feature extraction of tool wear. Hou et al. [20] combined channel attention with multiscale convolution to extract multiscale spatial-temporal features in cutting signals for tool wear monitoring. Zhou et al. [21] proposed Dual Attention Mechanism Network to learn pixel feature dependency and inter-channel correlation respectively. He et al. [22] proposed a cross-domain adaptation network based on attention mechanism to realize the tool wear state recognition and prediction. Dong et al. [23] combined the channel attention mechanisms of CaAt1 and CaAt5 with ResNet18 for tool wear monitoring. Guo et al. [24] focused on the important parts of the sequence information through an attention mechanism to achieve a multi-step tool wear prediction. Lai et al. [25] used the attention mechanism for weighted fusion of frequency and spatial features and performed an interpretability analysis of tool wear state recognition algorithms. Feng et al. [26] captured the complex spatial-temporal relationship between tool wear and features by weighting features in both the spatial and temporal dimensions through an attentional mechanism. Huang et al. [27] fused different scale features extracted by CNN through an attention mechanism to achieve tool wear prediction from multi-sensor data.

Three attention modules were mentioned in the above study. The first attention module type is a weighted fusion of the results of different layers of networks. By fusing different networks it is true that the recognition accuracy can be improved, but it also increases the overall computational cost of the network. The second type of attention module is the conversion of a temporal signal into a 2D time-frequency map by time-frequency transformations, and to perform feature extraction by channel and spatial attention mechanism in the field of image classification. However, the data structure of images is more complex compared to sequences and requires a network with larger number of parameters. The last attention module type is a channel attention mechanism for multidimensional sequences. The weighted representation of features is achieved by learning the weights of each channel in the feature map. However, current researchers mainly realize the channel attention by squeezing excitation on the channels of multidimensional sequences, which is inefficient in capturing the dependencies between the channels in this mechanism [28].

The attention operation is realized with 1D convolution instead of fully connected squeezing excitation. The coverage of cross-channel interactions is controlled by the size of the 1D convolutional kernel, which improves the inter-channel dependencies. The main contributions of this paper are as follows:(1)A 1DCCA-CNN model was proposed to realize the tool wear state recognition. The features of the cutting signal were extracted by a one-dimensional convolutional neural network. A novel channel attention was proposed. The inter-channel weight relationships are learned by one-dimensional convolution rather than squeezed excitation of fully connected layers, which can improve the interaction ability between different channels, and extract the features strongly related to the tool wear to improve model performance.(2)Validation was performed on the PHM2010 public cutting dataset. Cross-validation datasets with different groups were designed. The proposed model’s performance on the cross-validation dataset was evaluated by confusion matrix, accuracy, precision and recall. The superiority of the proposed model was verified by comparing with other models.

## 2. Structure of the Proposed Model

### 2.1. Overall Framework

The overall framework consists of data processing and 1DCCA-CNN model (Figure 1). The PHM2010 public cutting dataset [29] was used for model training and validation. The time-domain features (max, mean and variance) of the raw signals in the PHM2010 cutting dataset were extracted to form the training dataset and test dataset. The proposed 1DCCA-CNN composed of three convolutional layers, a channel attention layer and fully connected layers. The first convolutional layer was used to recognize low-level features and the second convolutional layer was used to recognize mid-level features. At this time, the channel dimension of the feature map was large, and a channel attention layer needed to be accessed. This channel attention layer adopted 1DCNN to realize the calculation of channel weights, which can improve the information interaction between channels. The third convolutional layer was used for extract high-level features. Finally, the feature map was mapped to the category dimension by a fully connected layer to realize the tool wear state recognition.

### 2.2. 1D Convolutional Neural Network Layer

1D Convolutional Neural Network (1D CNN) is a neural network architecture for processing one-dimensional sequential data in deep learning, which has the advantages of localized feature extraction, translational invariance and hierarchical feature extraction. In the tool wear state recognition task, some of the key features would be distributed in the local area, therefore, the superior local feature extraction capability of convolutional neural can effectively capture these features. The translational invariance of the convolutional layer can also automatically learn the tool wear characteristics at different positions. Meanwhile, the bottom convolutional layer of the convolutional neural network can capture low-level features, and the high-level convolutional layer can capture deeper abstract features, thus improving tool wear state recognition accuracy. There are a total of three 1D CNN layers in this study which are all composed of convolution operation, activation function, pooling layer and batch normalization (Figure 2).

1D convolutional operations are used to capture localized features in the input sequence and are applicable to a variety of sequence data such as text, audio, and time series. The convolution kernel is slid over the input sequence and the dot product of the convolution kernel with the input is computed to extract features at different locations (Figure 3). This allows the model to automatically capture localized features in the input sequence without explicitly specifying the location of the features. The formula for the convolution operation is shown in Equation (1).
(1)$$y_{i}=\overset{M-1}{\underset{j=0}{\sum }}x_{i+j}\times h_{j}$$
where $y_{i}$ denotes the ith element of the output sequence of the convolution operation, $i=0,\;1,\ldots ,\;\left(N-M\right)$, M is the convolution kernel size, and N is the length of the input sequence; $x_{i+j}$ is the i+jth element in the input sequence x; and $h_{j}$ is the jth weight in the convolution kernel h.

The pooling operation downsamples the feature map to reduce the size. The pooling operation also extracts the most salient features, reduces the number of parameters in the model, and helps prevent overfitting. The maximum pooling layer was used in this study as shown in Equation (2): (2)$$y_{j}=\max\left(a\left[j:j+K\right]\right)$$
where $y_{i}$ denotes the ith element of the output sequence of the pooling operation, $i=0,\;1,\ldots ,\frac{N}{K}$, K is pooling kernel size, N is the length of the input sequence; $a\left[j:j+K\right]$ is the K consecutive elements from the input sequence a, j=0, 1,…, N−K.

The activation function introduces nonlinear properties that allow neural networks to learn and represent more complex functional relationships. Among them, RELU is widely used in convolutional neural networks due to its high computational efficiency while mitigating the gradient vanishing problem and its contribution to the enhancement of the generalization ability of the model. The formula of RELU is shown in Equation (3).
(3)$$\mathrm{RELU}\left(x\right)=\left\{\begin{array}{c} 0,o\;x<0\\ x,o\;x\geq 0 \end{array}\right.$$
where x is a point in the feature sequence.

The main role of Batch Normalization (BN) is to accelerate the training of neural networks, to improve the stability of the training, as well as to reduce the gradient vanishing problem. The BN contributes to the training of the network by normalizing each small batch of data to adjust the distribution of the input data to a standard normal distribution with a mean of zero and a standard deviation of one. For input data with batch size m and channel size C, the calculation process of BN is as follows:

Step 1: Calculate the sample mean for each channel c
(4)$$\mu_{c}=\frac{1}{m}\overset{m}{\underset{i=1}{\sum }}x_{i}^{c}$$
where $x_{i}^{c}$ is the cth channel of the ith sample.

Step 2: Calculate the standard deviation of each channel c
(5)$$\sigma_{c}=\sqrt{\frac{1}{m}\overset{m}{\underset{i=1}{\sum }}{\left(x_{i}^{c}-\mu_{c}\right)}^{2}+\epsilon }$$
where $x_{i}^{c}$ is the cth channel of the ith sample; ϵ is the smoothing term, which a very small positive number that prevents the divisor from going to zero, and is here set to 10^−5^. 

Step 3: Normalize for each channel c
(6)$${\hat{x}}_{i}^{c}=\frac{x_{i}^{c}-\mu_{c}}{\sigma_{c}}$$

Step 4: Perform a linear transformation on the normalized values
(7)$$y_{i}^{c}=\gamma {\hat{x}}_{i}^{c}+\beta$$
where $y_{i}^{c}$ is the normalized result of the final output, γ is the learnable scaling factor, and β is the learnable translation factor.

### 2.3. 1D Channel Attention

The main significance of the channel attention is to enhance the model’s adaptability to different channel features. By automatically adjusting the weight of each channel, the model can better capture key features, reduce redundant information, and improve model performance and generalization ability. The channel attention mechanism traditionally applied to multidimensional sequential tasks is realized by squeezing excitation through a fully connected layer (Figure 4). This study takes advantage of the good cross-channel interaction capability that convolution has, and proposes to use 1DCNN instead of the commonly used squeezing excitation of the fully connected layer to realize the channel attention mechanism (Figure 4), which can enhance the information interaction capability between channels. For the input sequence of L×C, the implementation process of the 1DCNN attentional mechanism is as follows:

Step 1: Perform one-dimensional global average pooling on the input sequence
(8)$${\mathrm{GAP}}_{c}=\frac{1}{L}\overset{L}{\underset{i=1}{\sum }}x_{i,\;\;\;c}$$
where $x_{i,\;c}$ is the eigenvalue of channel c of the input sequence at position i, c=0, 1,…,C, C is the channel size of the input sequence, and L is the length of the input sequence.

Step 2: Perform one-dimensional convolution operation on the features in the channel dimension
(9)$$y_{i}=\overset{M-1}{\underset{j=0}{\sum }}c_{i+j}\times h_{j}$$
where $y_{i}$ denotes the ith element of the output sequence of the convolution operation, i=0, 1,…, C, C is the channel size of the input features, M is convolution kernel size; $c_{i+j}$ is the eigenvalue on the i+jth channel in the feature c; $h_{j}$ is the jth weight in the convolution kernel h.

Step 3: Sigmod calculations are performed on the results of the one-dimensional convolution to obtain the weights of each channel
(10)$$\sigma \left(y\right)=\frac{1}{1+e^{-y}}$$
where y is the sequence of features after one-dimensional convolution.

Step 4: The complete channel attention is realized after loading the obtained weights on each channel.

### 2.4. Full Connected Layer

The fully connected layer integrates the feature information extracted from the previous convolutional layers in the convolutional neural network into a global feature vector and is used in the final classification or regression task. The neurons in the fully connected layer are connected to all the neurons in the previous layer, and by learning the weight parameters, the nonlinear relationships in the data can be captured and the expressive power of the network can be improved. In this study the fully connected layer consists of two linear mapping layers, a RELU activation function and a Sigmod (Figure 5). Linear mapping layer 1 maps global feature vectors to lower dimensions. The RELU activation layer implements the nonlinearization. Linear mapping layer 2 maps global feature vectors to the classification dimension. Sigmod maps the feature values between 0 and 1 to get a score for each category to realize the tool wear state recognition. The specific process is as follows:(11)$$Linear1\left(x\right)=xw_{1}+b_{1}$$
(12)$$Linear2\left(x\right)=RELU\left(Linear1\left(x\right)\right)w_{2}+b_{2}$$
(13)$$Out\left(x\right)=Sigmod\left(Linear2\left(x\right)\right)$$
where w is the weight matrix; b is the bias; x is the input vector; the formula for RELU is shown in Equation (3) and the formula for Sigmod is shown in Equation (10).

## 3. Experiment

### 3.1. Experiment Set

The PHM2010 public cutting dataset was used to validate the performance of the model. There were six sets of cutting experiments in the PHM2010 public cutting dataset for the full life cycle of the tool, but only three sets, C1, C4, and C6, gave the values of the tool wear after each tool travel. Therefore, only these three sets of experimental data were used. The cutting condition is shown in Figure 6. Roder Tech RFM769 high speed CNC machine was used for the experiment. The cutting work piece was stainless steel. The tool used was 3-flute ball cutters. The cutting mode was dry milling and side milling. The length of each pass was 108 mm. The cutting parameters are shown in Table 1.

Cutting forces were measured in three directions during cutting using a Kistler quartz 3-component platform dynamometer. Three Kistler piezoelectric accelerometers were also deployed to measure triaxial vibration of the workpiece. The Kistler acoustic emission sensor was used to measure the high-frequency stress waves generated during the cutting process. The data collected by these three sensors were unified by the NI data acquisition card to form a 7-channel signal, which were x-direction cutting force, y-direction cutting force, z-direction cutting force, x-direction vibration, y-direction vibration, z-direction vibration and acoustic emission signal. After each pass, the average wear bandwidth of the flank face VB of each insert was measured using the LEIC MZ12 Microscope as the wear value of the insert. In order to intuitively illustrate the areas of tool wear measured in the dataset, The tool image taken at other factory was added to the Off-line measurement section of Figure 6. The average wear value of the three inserts was used as the wear value of the whole tool. The full life cycle tool wear for the three sets of experiments is shown in Figure 7.

### 3.2. Data Preparation

Tool wear can be divided into initial wear, normal wear and severe wear according to the rate of change of wear. In the early stage of tool use, the surface of the newly sharpened tool is relatively rough. At this time, the contact stress is high and the wear is faster, this stage is defined as the initial wear of the tool. With the increase in cutting time, the cutting process tends to stabilize, the wear is flat, this stage is defined as the normal wear stage of the tool. When the tool becomes blunt with the increase of use time, the cutting performance of the tool decreases sharply and the wear accelerates again, this stage is defined as the severe molding stage of the tool. According to the slope of the wear curve of the full life cycle of the tool in Figure 7, the tool was divided into three wear stages. The 1st pass to the 50th pass was defined as initial wear, the 51st pass to the 175th pass was defined as normal wear, and the 176th pass to the 315th pass was defined as severe wear.

As shown in Figure 8, we preprocessed the raw signals collected by the sensors. The raw signal consisted of a total of seven channels. A non-overlapping window was divided for each channel, and then time-domain features were extracted for the signals in each window, which were the maximum value, the mean value and the variance. The three extracted features were then connected in the temporal direction to form a new input sequence of dimension ℝ100×21. Due to the different units and value ranges of data collected by different sensors, Min-Max normalization was used to eliminate the difference in magnitude between different channels in order to avoid the model being more sensitive to certain features and to accelerate the convergence of the model. The labels of the sequences are defined for training based on the fact that they are in the wear stage. Due to the uneven amount of data under the three types of tool wear, especially the normal and severe wear stages were much larger than the initial wear, which would l cause a reduction in the generalization ability of the model. Therefore, the raw signals of the initial wear stage were upsampled by repartitioning the window randomly. The normal and severe wear stage samples were randomly selected for downsampling. Finally, the sample size of each wear stage for each of the three sets of experiments, C1, C4 and C6, was one thousand, totaling three thousand samples for each set of experiments. To better validate the performance of the proposed model, three sets of crossover datasets divided by experimental groups were designed. The three sets of crossover datasets are shown in Table 2.

### 3.3. Model Parameters and Training Parameter Settings

Training was performed in Windows 10 platform using GPU. The graphics card was RTX 3090 and the CPU was AMD EPYC 7624. The deep learning network was built by Pytorch 1.11.0 framework. Python version was 3.8. Cuda version was 11.3. Adam was used as the optimizer for training, and improved the generalization of the model by setting weight_decay to prevent overfitting in model training. The loss function was calculated using the cross-entropy function. The parameters of each layer of the proposed model are shown in Table 3. The hyperparameters for model training are shown in Table 4. The training dataset loss curve and test dataset accuracy curve of the proposed model during training are shown in Figure 9. It was observed that the model started to converge gradually at the 40th round of training, and the accuracy of the test dataset quickly reached a high level.

### 3.4. Analysis of Experimental Results

The confusion matrix of the recognition results of the proposed 1DCCA-CNN model on the three datasets is shown in Figure 10. To further evaluate the model’s performance, Accuracy, Precision, and Recall were used as metrics. The formulas for calculating the three metrics are as shown in Equations (13)–(15), respectively. Accuracy measures the accuracy of the model on the overall dataset. Precision focuses on the accuracy of the positive case prediction. Recall focuses on the ability of the model to recognize positive cases. In the tool wear recognition task, the initial and normal wear stages should avoid false alarms leading to early tool change and waste, while in the severe wear stage should try to recognize the severe wear state in time. Therefore, more attention should be paid to the model’s precision in the initial and normal wear phases, and more attention should be paid to the model’s recall in the severe wear phase. The proposed model’s evaluation metrics for the three datasets are presented in Table 5.
(14)$$Accuracy=\frac{TP+TN}{TP+TN+FP+FN}$$
(15)$$Precision=\frac{TP}{TP+FP}$$
(16)$$Recall=\frac{TP}{TP+FN}$$
where $TP\;\left(True\;Positives\right)$ represents the number of positive samples properly classified as positive by the model; $TN\;\left(True\;Negatives\right)$ represents the number of negative samples properly classified as negative by the model; $FP\;\left(False\;Positives\right)$ represents the number of negative samples incorrectly classified as positive by the model; $FN\;\left(False\;Negatives\right)$ represents the number of positive samples incorrectly classified as negative by the model.

To better validate the properties of the model, two ablation models were set up for comparative validation. The first ablation model (CNN) removed the attention mechanism and only preserved the one-dimensional convolutional module compared to the proposed model. The second ablation model (SECA-CNN) uses squeeze excitation channel attention mechanism instead of 1DCNN channel attention in the proposed model. It was also compared with existing tool wear state recognition algorithms that were also validated using the T dataset. Dong et al. [23] used the attention mechanism in CaAt1 and CaAt5 for tool wear state recognition in ResNet-1d. Yin et al. [30] implemented 1D-CNN with DGCCA for tool wear state recognition. Li et al. [31] combined GBDT with H-ClassRBM for tool wear state recognition. The comparison results of the recognition accuracy of different models are shown in Table 6.

It can be seen that the recognition accuracies of the proposed model under the three datasets were improved by 3.8%, 6.7%, and 4.7%, respectively. Compared with the CNN without channel attention, which verified the significance of the attention mechanism in optimizing the quality of the extracted features and improving the performance of the tool wear state recognition model. Compared with SECA-CNN, the recognition accuracy of the proposed 1DCCA-CNN improved by 1.2%, 2.1%, and 1.6% on the three datasets, respectively, which was due to the poor ability of the conventional squeeze excitation channel attention mechanism to capture channel dependencies, and resulted in the model’s generalization capability decrease. Using 1D CNN instead of squeeze excitation can effectively achieve inter-channel interaction. Compared to existing research findings, the proposed model demonstrated a significant lead in recognition accuracy on the T1 and T3 datasets. The recognition accuracy on the T1 dataset improved by 4% compared to the highest level achieved in existing research, while on the T3 dataset, the recognition accuracy improved by 5% compared to the highest level achieved in existing research.

## 4. Conclusions and Future Work

This paper proposed a tool wear status recognition algorithm based on 1DCNN channel attention mechanism. The low and intermediate features in the signal were extracted by two one-dimensional convolution layers, and the channels of the features were weighted by the proposed channel attention mechanism to enhance the important channels and capture the key features. Then the high-level features were fetched by the last one-dimensional convolutional layer, and finally mapped to the classification layer through the fully connected layer. The main contributions of this study are as follows:(1)The channel attention mechanism was proposed by using 1DCNN instead of the traditional squeeze excitation. The good cross-channel information acquisition ability of convolutional was used to improve the information interaction between channels, so as to effectively capture the dependency between channels.(2)The model performance was verified on PHM2010 public dataset. Compared with CNN without attention mechanism, the recognition accuracy of the proposed 1DCCA-CNN improved 3.8%, 6.7%, and 4.7% respectively under the three datasets, which verified the importance of applying channel attention to model performance improvement. Compared with SECA-CNN, which used the traditional squeeze excitation attention mechanism, the recognition accuracy of the proposed 1DCCA-CNN improved 1.2%, 2.1%, and 1.6%, respectively, under the three datasets, which verified the superior performance of the proposed 1DCNN attention mechanism.(3)The proposed model had high accuracy, precision and recall, which verified the recognition performance of the model. Compared with the existing research results, the proposed model performed well on the T1 and T3 datasets, increasing by 4% and 5% respectively compared with the highest level of the existing research results, which verified the superior properties of the model.

The proposed methodology still has some limitations. First of all, the recognition accuracy of the model on T2 datasets is still slightly lower than the highest level of existing research results, and further improved performance will be considered in data preprocessing and network depth in the future. Secondly, this study is only applicable to a single working condition, and the tool wear state recognition algorithm under multiple working conditions should be studied later. Finally, the explainability of the proposed model is poor, and the visualization of the model should be further improved in the future to improve the explainability.

## Figures and Tables

**Figure 1 micromachines-14-01983-f001:**
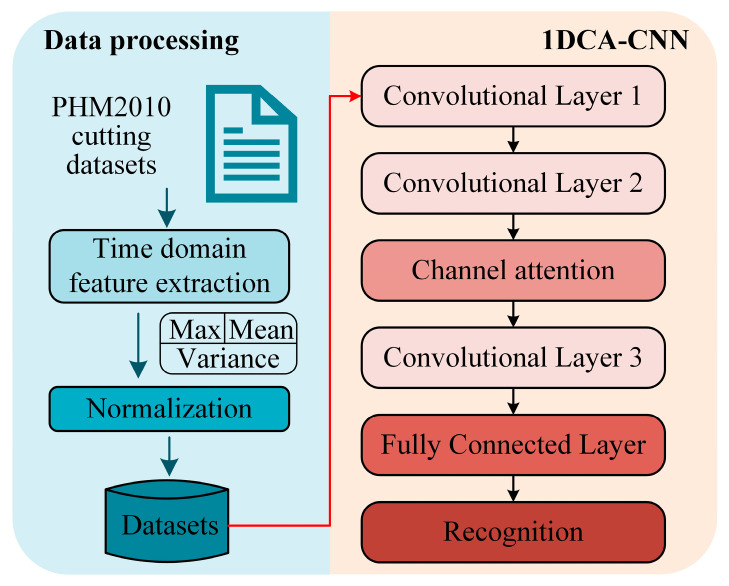
The overall structure of the proposed model.

**Figure 2 micromachines-14-01983-f002:**
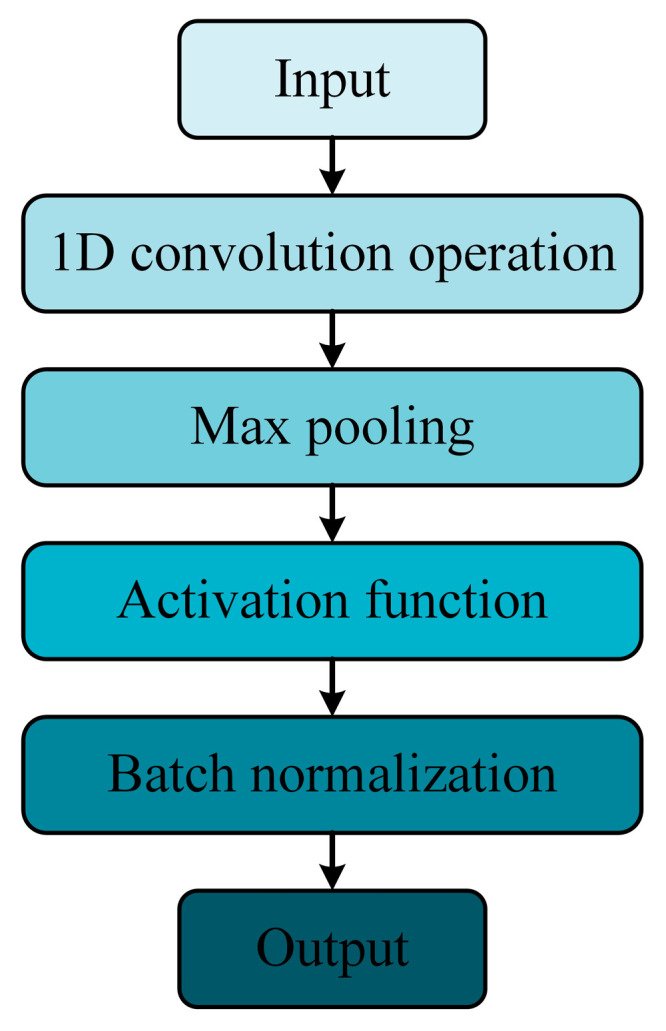
The structure of a 1D convolutional layer.

**Figure 3 micromachines-14-01983-f003:**
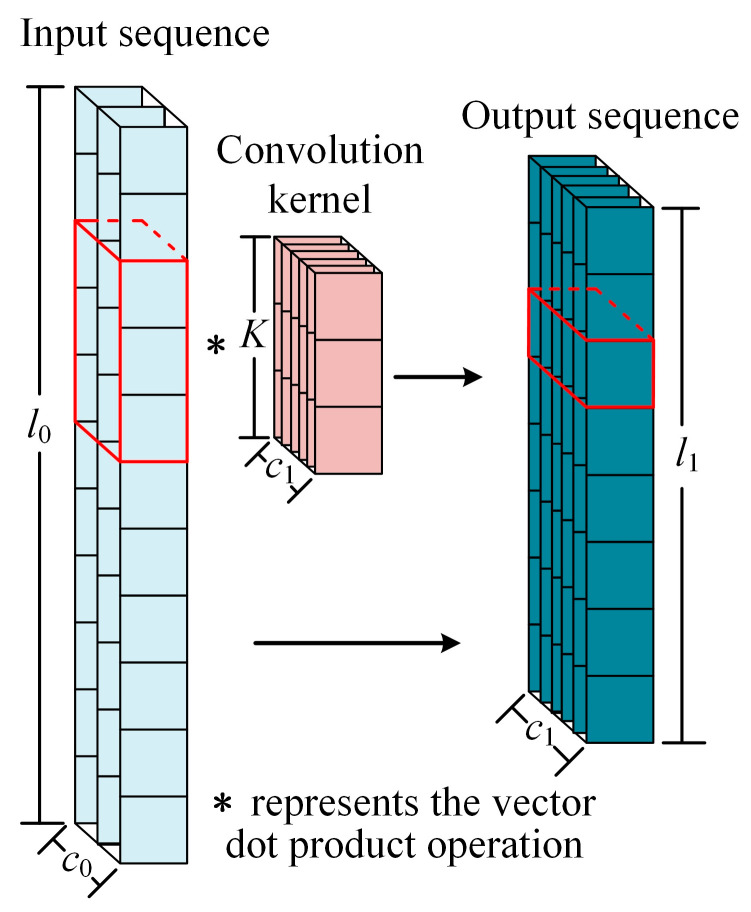
The operation of one-dimensional convolution.

**Figure 4 micromachines-14-01983-f004:**
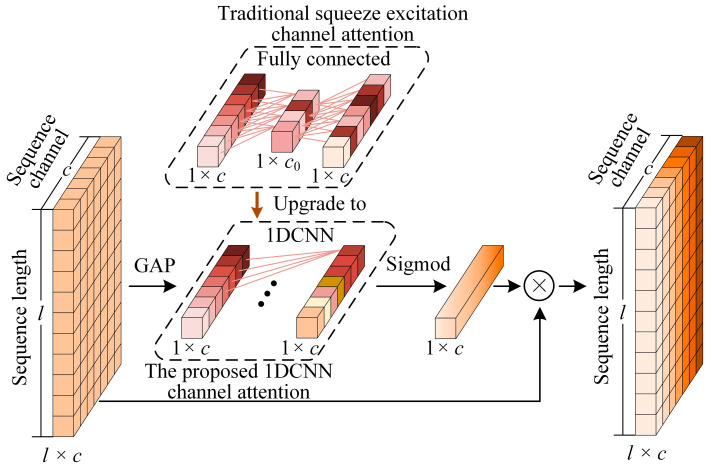
Traditional squeeze excitation channel attention and the proposed structure of a channel attention mechanism based on 1DCNN.

**Figure 5 micromachines-14-01983-f005:**
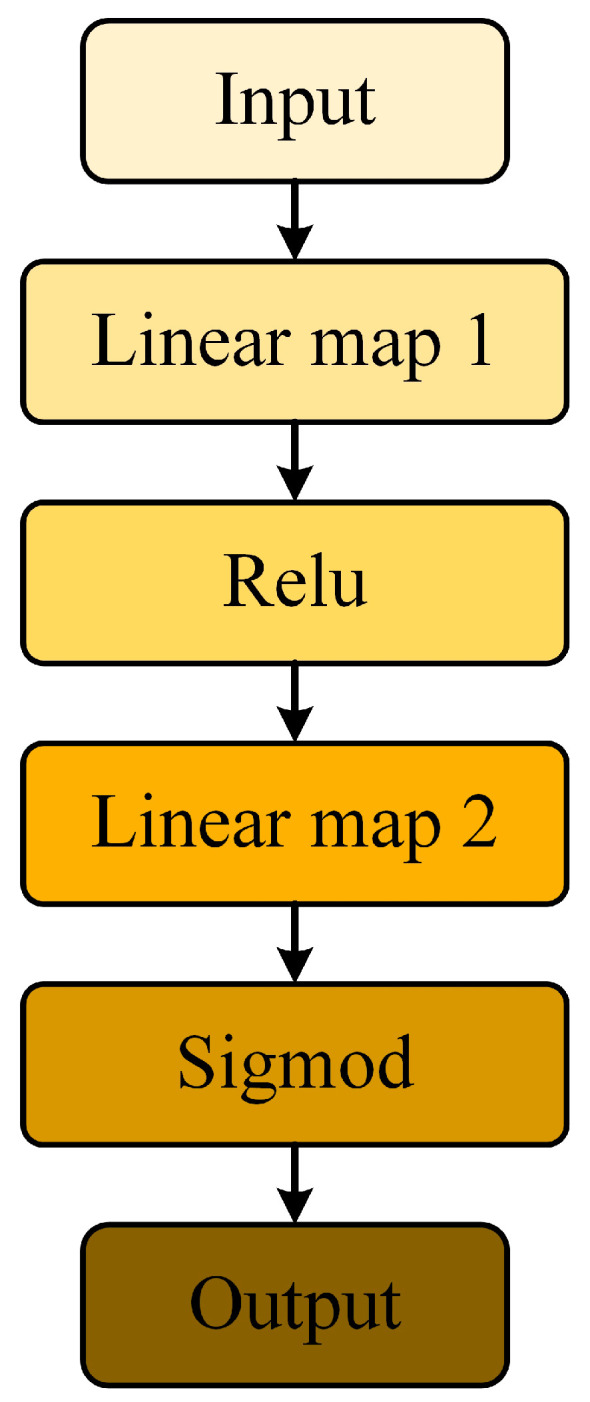
The structure of the fully connected layer.

**Figure 6 micromachines-14-01983-f006:**
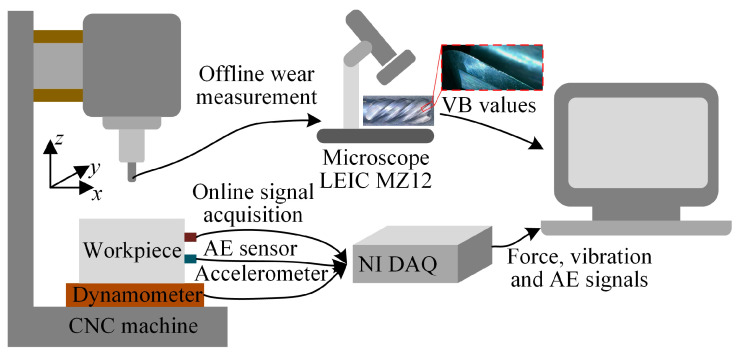
PHM2010 public cutting dataset working conditions.

**Figure 7 micromachines-14-01983-f007:**
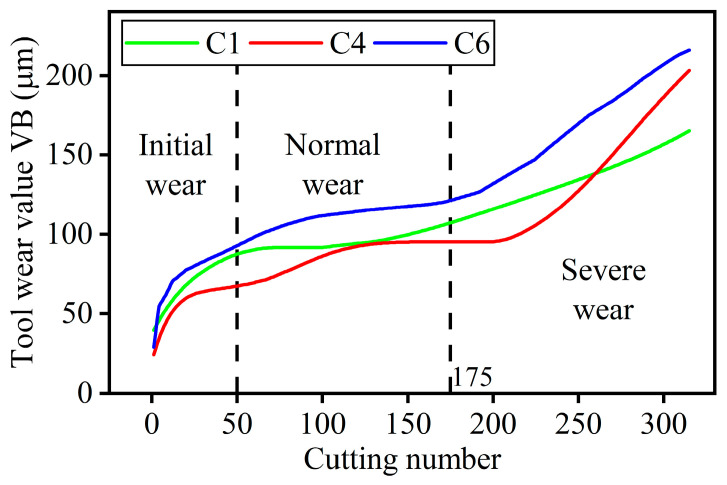
Tool Full life cycle wear curve for in PHM2010 dataset.

**Figure 8 micromachines-14-01983-f008:**
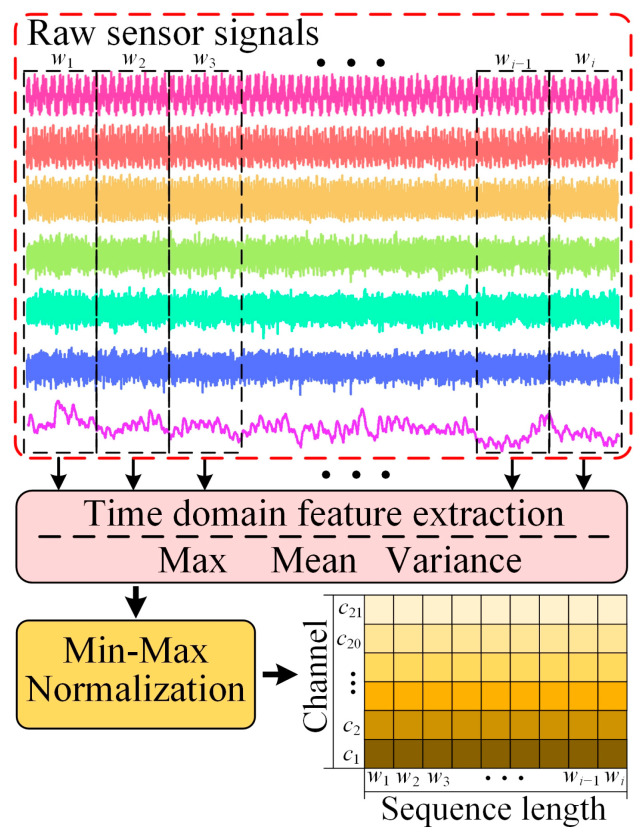
The flow of raw sensor data processing.

**Figure 9 micromachines-14-01983-f009:**
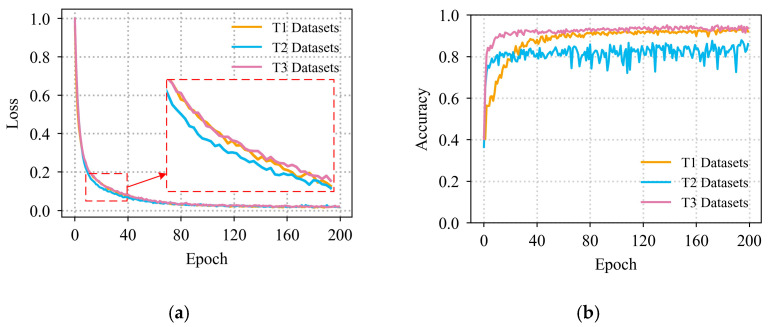
The proposed model training situation (**a**) Loss curves on the training dataset (**b**); Accuracy curve on the test dataset.

**Figure 10 micromachines-14-01983-f010:**
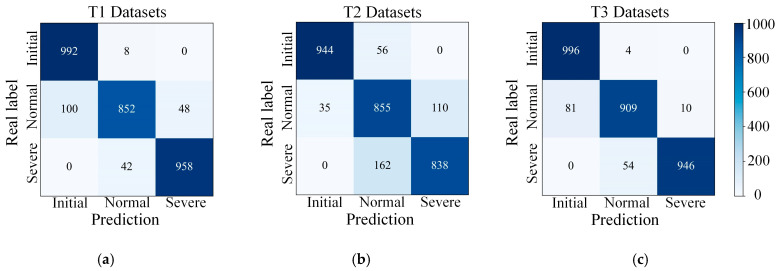
The confusion matrix of the proposed model on the (**a**) T1 test datasets, (**b**) T2 test datasets, and (**c**) T3 test datasets.

**Table 1 micromachines-14-01983-t001:** Experimental parameters for the PHM2010 public cutting dataset.

Experimental Parameters	Size
Spindle speed (RPM)	10,400
Feed rate (mm/min)	1555
Cutting depths (mm)	0.2
Cutting width (mm)	0.125
Sampling frequency	50 kHz

**Table 2 micromachines-14-01983-t002:** Three cross-validation datasets.

Datasets	Training Datasets	Test Datasets
T1	C1 + C4	C6
T2	C4 + C6	C1
T3	C6 + C1	C4

**Table 3 micromachines-14-01983-t003:** The parameters of each layer of the proposed model.

Layer Name	Description	Parameters Setting
Convolutional layer 1	1D Convolution	filters = 42, kernel size = 3, padding = 0, stride = 1, activation = RELU, out size = 98 × 42
	1D Max pooling	pool size = 2, padding = 0, stride = 2, out size = 49 × 42
	Batch normalization	feature number = 42, out size = 49 × 42
Convolutional layer 2	1D Convolution	filters = 84, kernel size = 3, padding = 0, stride = 1, activation = RELU, out size = 47 × 84
	Max pooling	pool size = 2, padding = 0, stride = 2, out size = 23 × 84
	Batch normalization	feature number = 84, out size = 23 × 84
Channel attention	GMP	out size = 1 × 84
	1D Convolution	filters = 1, kernel size = 3, padding = 1, stride = 1, out size = 1 × 84
	Sigmod	out size = 1 × 84
	Weighted multiplication	out size = 23 × 84
Convolutional layer 3	1D Convolution	filters = 168, kernel size = 3, padding = 0, stride = 1, activation = RELU, out size = 21 × 42
	1D Max pooling	pool size = 2, padding = 0, stride = 2, out size = 10 × 168
	Batch normalization	feature number = 168, out size = 10 × 168
Fully connected layer	Flatten	out size = 1 × 1680
	Linear 1	out size = 1 × 64
	RELU	out size = 1 × 64
	Linear 2	out size = 1 × 3
	Sigmod	out size = 1 × 3

**Table 4 micromachines-14-01983-t004:** Hyperparameters for model training.

Hyperparameters	Size
Learning rate	0.0001
Batch size	256
Epoch	200
Weight_decay	0.05

**Table 5 micromachines-14-01983-t005:** Evaluation metrics for the proposed model in the three datasets.

Numerical (%)		Evaluation Metrics		
		Precision	Recall	Accuracy
T1 datasets	Initial wear	90.8	99.2	93.4
	Normal wear	94.5	85.2	
	Severe wear	95.2	95.8	
T2 datasets	Initial wear	96.4	94.4	87.9
	Normal wear	79.7	85.5	
	Severe wear	88.4	83.8	
T3 datasets	Initial wear	92.5	99.6	95.0
	Normal wear	94.0	90.9	
	Severe wear	99.0	94.6	

**Table 6 micromachines-14-01983-t006:** The recognition accuracies of different comparative models.

Model	T1 Datasets	T2 Datasets	T3 Datasets
CNN	89.7	81.2	90.3
SECA-CNN	92.3	85.8	93.4
CaAt-ResNet-1d [23]	86.0	88.0	89.2
1D-CNN-DGCCA [30]	89.5	88.4	90.0
GHCRBM [31]	89.2	90.8	58.4
The proposed model	93.5	87.9	95.0

## Data Availability

Not applicable.

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
