# Peer review of "Tool Wear State Recognition Based on One-Dimensional Convolutional Channel Attention"

_micromachines, 2023, doi:10.3390/mi14111983_

Round 1

Reviewer 1 Report

This paper proposed a novel channel attention mechanism based on 1DCNN, which achieved better results in the tool wear state recognition task. The information interaction between the channels was enhanced by 1DCNN, which is interesting. I recommend the following revisions are properly made: 

1) The milling experiment is face milling or side milling?

2) Where is the specific location of tool wear in the paper? Bottom or side? Please provide some photos of the tool wear area.

3) How to guarantee the generalization ability of the proposed network?

4) It is noted that the PHM2010 public datasets is based on a single experiment condition. I wander how the authors verify the model accuracy based on this limited dataset, even though the data volume could be huge. The testing experimental condition is the same as that of the training dataset.

5) Please, check if all the symbols used in the various equations are well-defined in the text.

6) From the Fig. 7, the tool wear shows a clear piece-wise linear relationship with cutting time, and a regression may reach that goal. Is such a complex deep network necessary for the 3-state classification?

The quality of English language is well.

Reviewer 2 Report

Dear authors,

firstly, this is this is good work considering how difficult it is to do online tool wear monitoring. The paper also covers an interesting topic. In order to improve the quality of paper and better understanding, you should read the following questions and suggestions. Therefore, you should thoroughly review and revise your work.

1. You claim that RELU is widely used in Convolutional Neural Networks. Do you have sources on the basis of which you concluded that?

2. In Equation 5, there is a very small positive number that prevents division to zero. When you take a formula that is not the right standard deviation of the formula, can you get inadequate results? Specify, in what case and how often can this happen?

3. Did you have one set of cutting parameters and based on that, did the measurement? Did you use a different combinations of cutting parameters? Please, clarify that. Why different combinations of parameters were not used?

4. Why only three sets, C1, C4, and C6, gave the values of the wear after each tool travel? What happened to the others, why were they not used? What do you mean by “experimental set”? Please present it more clearly.

5. Why did you named “server” wear? Change the name to “catastrophic” or “intensive” wear. Consult the literature in the field.

6. In figure 7, what is cutting pass? Cutting time usually goes to that axis. Please describe what one pass means. And how long it lasts, how long pass are, or whatever.

7. Accuracy, Precision, and Recall were used as metrics. Why exactly did you take the presented formulation? Do you have a foothold in previous literature for it?

Best regards
